# Overexpression of *SgGH3.1* from Fine-Stem Stylo (*Stylosanthes guianensis* var. *intermedia*) Enhances Chilling and Cold Tolerance in *Arabidopsis thaliana*

**DOI:** 10.3390/genes12091367

**Published:** 2021-08-31

**Authors:** Ming Jiang, Long-Long Ma, Huai-An Huang, Shan-Wen Ke, Chun-Sheng Gui, Xin-Yi Ning, Xiang-Qian Zhang, Tian-Xiu Zhong, Xin-Ming Xie, Shu Chen

**Affiliations:** 1Department of Grassland Science, College of Forestry and Landscape Architecture, South China Agricultural University, Guangzhou 510642, China; mingjiang@stu.scau.edu.cn (M.J.); malonglong@stu.scau.edu.cn (L.-L.M.); wingann@126.com (H.-A.H.); gui8116@stu.scau.edu.cn (C.-S.G.); nxyiii@stu.njau.edu.cn (X.-Y.N.); maizerice@scau.edu.cn (X.-Q.Z.); zhongxinbi@scau.edu.cn (T.-X.Z.); 2Guangdong Engineering Research Center for Grassland Science, Guangzhou 510642, China; 3Gansu Engineering Laboratory of Applied Mycology, Hexi University, Zhangye 734000, China; ksw0705@126.com; 4Department of Ornamental Horticulture, College of Horticulture, Nanjing Agriculture University, Nanjing 210095, China

**Keywords:** Gretchen-Hagen 3, *Stylosanthes*, overexpression, transgenic, cold tolerance, auxin

## Abstract

*Stylosanthes* (stylo) species are commercially significant tropical and subtropical forage and pasture legumes that are vulnerable to chilling and frost. However, little is known about the molecular mechanisms behind stylos’ responses to low temperature stress. Gretchen-Hagen 3 (GH3) proteins have been extensively investigated in many plant species for their roles in auxin homeostasis and abiotic stress responses, but none have been reported in stylos. *SgGH3.1*, a cold-responsive gene identified in a whole transcriptome profiling study of fine-stem stylo (*S. guianensis* var. *intermedia*) was further investigated for its involvement in cold stress tolerance. *SgGH3.1* shared a high percentage of identity with 14 leguminous GH3 proteins, ranging from 79% to 93%. Phylogenetic analysis classified SgGH3.1 into Group Ⅱ of GH3 family, which have been proven to involve with auxins conjugation. Expression profiling revealed that *SgGH3.1* responded rapidly to cold stress in stylo leaves. Overexpression of *SgGH3.1* in *Arabidopsis thaliana* altered sensitivity to exogenous IAA, up-regulated transcription of *AtCBF1-3* genes, activated physiological responses against cold stress, and enhanced chilling and cold tolerances. This is the first report of a *GH3* gene in stylos, which not only validated its function in IAA homeostasis and cold responses, but also gave insight into breeding of cold-tolerant stylos.

## 1. Introduction

Species of *Stylosanthes* (stylo) are among the most economically important forage and pasture legumes in tropical and subtropical regions of Asia, Africa, South America, and Australia [1]. Since stylos were first introduced to South-East Asia in 1949, *S. guianensis* has become the most productive and widely adapted stylo species for the acid infertile soils of humid and subhumid Asia. In China, *S. guianensis* is mainly used as a cover crop to suppress weed growth in plantations as well as a green manure to improve soil fertility and nitrogen, and then freshly cut and processed into leaf meal to feed animals [2]. *S. guianensis* exhibits high tolerance to drought and soil infertility, but are very vulnerable to frost and chilling stress [3,4], which makes it difficult for most stylo cultivars to survive through winter even in subtropical areas such as Guangzhou (113°21′50′′, 23°10′10′′) (Appendix A). Fine-stem stylo (*S. guianensis* var. *intermedia*) is a variety of *S. guianensis*, but it is morphologically and physiologically very different from the other varieties of this species. It is a much smaller plant with maximum height of 30–40 cm and stem diameter < 3 mm, compared with common stylos which usually have a maximum height over 100 cm and old stems highly coarse and woody. More importantly, fine-stem stylo has much better frost tolerance than the common stylos and can survive at temperature as low as −10 °C [5]. Thus, understanding molecular mechanisms associated with cold responses and tolerance in fine-stem stylo could aid in the development of cold-tolerant cultivars capable of surviving subtropical winters.

Temperate plants have evolved a mechanism called cold acclimation that allows them to enhance their freezing tolerance through pre-exposure to non-freezing low temperature. Cold acclimation requires the coordination of transcriptional, biochemical, and physiological changes. C-repeat binding factors/dehydration-responsive protein-binding factors (CBFs/DREBs) activate cold-responsive (*COR*) genes during cold acclimation, resulting in the accumulation of cryoprotectants and the acquisition of freezing tolerance [6,7]. In this process, adjustments in membrane composition are made to avoid damage caused by freezing temperatures, stress-related proteins and sugars are accumulated to avoid dehydration caused by the immobilization of water around ice nuclei, antioxidant enzymes are activated, and the cold-sensitive photosynthetic machinery is protected [8,9]. In addition, transcript profiling indicated that CBF-regulated genes were engaged not only in the cold stress response, but also in hormone response, implying a connection between CBF-mediated cold acclimation and hormonal responses. [10,11].

A previous RNA-seq analysis was performed by our lab on fine stem-stylo leaves under cold stress, which revealed that a *Gretchen-Hagen 3.1* (*GH3.1*) homologous gene (*SgGH3.1*) was induced by low temperature (NCBI BioProject PRJNA316912). GH3 proteins were divided into three major clades according to their phylogenetic relationships, identified as Groups I, II, and III [12,13]. GH3 proteins of Group II have similar substrate preference, specifically conjugate auxins with amino acids in vitro and in vivo. Using mutants or transgenic lines with altered gene expression, it has been demonstrated that several Group II GH3 members are associated with responses to biotic and abiotic stresses in *A. thaliana*, *Oryza sativa* and *Citrus sinensis* [14,15,16,17,18,19]. These studies suggest that Group II GH3 proteins link regulation of plant growth and development with abiotic stress responses through mediating auxin homeostasis. Although it has been well established that auxin plays an important role in responses to abiotic stress, how IAA conjugation changes growth, development and defensive responses during abiotic stress remains largely unknown due to the complicated and extensive crosstalk between auxin homeostasis and abiotic stress responses.

To further investigate the role of *SgGH3.1* in cold responses in fine-stem stylo, expression variation of *SgGH3.1* gene was determined in stylos under cold stress, and transgenic *Arabidopsis* plants overexpressing *SgGH3.1* were generated using *Agrobacterium*-mediated transformation. Effects of *SgGH3.1* on cold tolerance and cold-responsive gene expression in transgenic plants were investigated. The results indicated that *SgGH3.1* responded to low temperature rapidly in stylo leaves and *SgGH3.1* overexpression changed IAA sensitivity, increased transcription of *AtCBF1-3* genes, triggered physiological responses to cold stress, and improved chilling and cold tolerance in *Arabidopsis*.

## 2. Materials and Methods

### 2.1. Gene Cloning and Bioinformatic Analysis

Total RNA was extracted from the leaf tissues of the fine-stem stylo variety ‘YueNong 01′ using TransZol Plant kit (TransGen Biotech, Beijing, China). The first cDNA was synthesized using TaKaRa PrimeScript II 1st Strand cDNA Synthesis Kit (Takara Biotechnology, Shiga, Japan). The full-length coding DNA sequences (CDSs) of *SgGH3.1* were amplified by PCR using a specially designed primer pair, *SgGH3.1*-pBA-F and *SgGH3.1*-pBA-R (Appendix A). PrimeSTAR^®^ Max DNA Polymerase (Takara Biotechnology, Japan) was used for the PCR. The 50 µL PCR reaction system contained 25 µL PrimeSTAR Max Premix (2X), 0.2 µM *SgGH3.1*-pBA-F, 0.2 µM *SgGH3.1*-pBA-R, and approximately 200 ng cDNA template. The PCR protocol consisted of 30 cycles of 20 s at 98 °C, 5 s at 55 °C and 30 s at 72 °C. The amino acid sequence of *SgGH3.1* was aligned with 20 *Arabidopsis* GH3 proteins (Appendix A) and 14 leguminous GH3.1 proteins (Appendix A) using T-coffee [20]. A phylogenetic tree was constructed from the alignment using MrBayes [21,22]. Percentage identities between the leguminous GH3.1 proteins were calculated based on the alignment.

### 2.2. Arabidopsis Transformation

Due to the low transformation efficiency and the long periodicity of callus induction and regeneration, generation of a transgenic stylo plantlet via *Agrobacterium*-mediated transformation remains extremely challenging. As a result, in this investigation, a model plant *A. thaliana* (*Arabidopsis*) was used to verify the function of *SgGH3.1*. *Arabidopsis* of Columbia ecotype (Col-0) was used for gene transformation. Plants were grown in a culture room at 22 °C with a relative humidity of 60% under short day conditions (12 h light and 12 h dark) with white light illumination (100 µmol photons·m^−2^·s^−1^). *SgGH3.1* was fused into an overexpression vector pBA002 carrying spectinomycin resistance gene and basta resistance (*bar*) gene, yielding a recombinant plasmid, pBA002-*SgGH3.1.* The recombination was conducted using T4 DNA ligase (Takara Biotechnology, Japan) after digestion with XbaI and BamHI. The *Agrobacterium tumefaciens* strain EHA105 carrying pBA002-*SgGH3.1* was used for *Arabidopsis* transformation using the floral dip method [23]. Seeds of *Arabidopsis* were collected and sown on Murashige and Skoog (MS) medium containing basta (5 mg/L) to select for transgenic plants (T1). The T1 plants were confirmed by amplification of *bar* gene from genomic DNA and amplification of *SgGH3.1* from cDNA. Homozygous third-generation (T3) seedlings were derived from self-crossed T2 plants exhibiting 1:0 segregation of basta resistance [24]. Independent T3 homozygous transgenic lines, *SgGH3.1*-OE1, 3 and 4 were used for further study.

### 2.3. IAA Treatment

Wild-type (WT) and T3 transgenic *Arabidopsis* lines, *SgGH3.1*-OE1, 3 and 4, were seeded on MS medium supplemented with 0, 2, 4, 6, 8 and 10 µM IAA and cultured at 22 °C under short day conditions (12 h light and 12 h dark) with white light illumination (100 µmol photons·m^−2^·s^−1^). Twenty seedlings from each treatment were collected for phenotypic measures after 18 days of cultivation.

### 2.4. Chilling Treatment

WT and T3 transgenic *Arabidopsis* lines, *SgGH3.1*-OE1 and 3, were seeded on MS medium and grown at 22 °C and 10 °C, respectively, under short day conditions (12 h light and 12 h dark) with white light illumination (60 µmol photons·m^−2^·s^−1^). Twenty seedlings from each treatment were collected for phenotypic measures after 20 days of cultivation.

### 2.5. Cold Treatment

Ten fine-stem stylo plants were vegetatively propagated by cuttings. Fresh shoots of 10 cm were cut from a single fine-stem stylo plant and cultured in water for rooting. After one week, rooted shoots were planted in pots (10 cm in height and 12 cm in diameter) containing soil mixtures of Jiffy^®^ substrate and vermiculite (3:1) and cultured in a growth chamber at 28 °C and 70% humidity under a 16/8-h (light/dark) photoperiod for four weeks. Tissues of root, stem and leaf were harvested and immediately immersed in liquid nitrogen for RNA extraction. Plants of uniform growth were selected for cold treatment at 4 °C and 70% humidity under a 16/8-h (light/dark) photoperiod. Leaf tissues were harvested at 0, 2, 6, 12, 24, and 48 h after cold treatment for relative gene expression analysis of *SgGH3.1*.

WT and T3 transgenic *Arabidopsis* lines, *SgGH3.1*-OE1 and 3, were seeded on MS medium and grown at 22 °C under short day conditions (12 h light and 12 h dark) with white light illumination (60 µmol photons·m^−2^·s^−1^). Three-week old plants were used for cold treatment at −4 °C under short day conditions (12 h light and 12 h dark) with white light illumination (60 µmol photons·m^−2^·s^−1^). Plants after an hour of cold treatment were sampled for relative gene expression analysis of *AtCBF1-3*. Fresh leaves after cold treatments of 0, 2, 6, 12, 24, and 48 h were harvested for physiological determination. Survival rates were assessed after 2 weeks of recovery at 22 °C following cold treatment.

### 2.6. Relative Gene Expression Analysis

Total RNA was extracted from nitrogen-frozen tissues using TransZol Plant Kit (TransGen Biotech, China). RNA concentration and purity was measured by a microplate spectrophotometer (BioTek Epoch, Winooski, VT, USA), and RNA integrity was assessed by 1% agarose gel electrophoresis. cDNA was synthesized with ReverTra Ace qPCR RT Master Mix with gDNA remover (Toyobo, Japan) following the manufacturer’s manual. To remove Genomic DNA, 8 µL reaction solution containing 0.5 µg RNA, 2 µL 4× DN Master Mix with gDNA remover was incubated at 37 °C for 5 min. 2 µL 5× RT Master Mix II was added to the reaction solution, and incubated at 37 °C for 15 min, 50 °C for 5 min and 98 °C for 5 min. The synthesized cDNA of 5-time dilution was used as template for Real-Time PCR by QuantStudio 3 Real-Time PCR System (Applied Biosystems, Forster City, CA, USA) with specific primer pairs (Appendix A). Real-time PCR was conducted using SYBR Premix Ex Taq II kit (Takara Biotechnology, Japan). The 25 µL Real-time PCR reaction solution contained 12.5 µL SYBR Premix Ex Taq II (2X), 0.2 µM forward primer, 0.2 µM reverse primer, and 1 µL cDNA template. The PCR protocol consisted of an initial pre-denaturation of 10 min at 95 °C, and 40 cycles of 15 s at 95 °C, 60 s at 60 °C and 30 s at 72 °C. The output data was analyzed using 2^−ΔΔCt^ method [25].

### 2.7. Physiological Determination

Fresh leaves after cold treatments of 0, 2, 6, 12, 24, and 48 h were harvested and used for determination of proline content [26], MDA content [27], soluble sugar content [26], relative electrolyte leakage and soluble protein content [26] following the published methods. Three replicates were conducted for each measurement.

### 2.8. Statistical Analysis

All morphological, physiological and Real-time PCR data was analyzed using the IBM SPSS Statistics 23.0 software. Significance of differences between samples or treatments were evaluated by Duncan’s multiple range test.

## 3. Results

### 3.1. SgGH3.1 Is a Member of GH3 Family

*SgGH3.1* consists of 602 amino acids. The phylogenic tree of *SgGH3.1* and *Arabidopsis* GH3.1 proteins indicates that *SgGH3.1* belongs to Group Ⅱ of GH3 family (Figure 1A). Sequence alignment shows that GH3.1 protein sequences are highly conserved among leguminous species. Percent identities between *SgGH3.1* and other leguminous GH3.1 proteins range from 79% to 93%. Among the 14 leguminous GH3.1 proteins, AhGH3.1 (*Arachis hypogaea*), AdGH3.1 (*A. duranensis*), and AiGH3.1 (*A. ipaensis*) are closest to *SgGH3.1*, showing percent identities of 91%, 93%, and 93%, respectively (Figure 1B). The phylogenetic analysis also clustered *SgGH3.1* and the three *Arachis* GH3.1 proteins into the same group, which is consistent with the plant taxonomy that *Stylosanthes* and *Arachis* belong to the same tribe Dalbergieae of subfamily Papilionoideae (Figure 1B).

### 3.2. SgGH3.1 Is Responsive to Cold Stress

Expression of *SgGH3.1* was observed in roots, stems, and leaves of stylo, and the expression was highest in roots, followed by stems and leaves (Figure 2A). To determine whether *SgGH3.1* is responsive to cold stress, stylo plants were subjected to cold treatment (4 °C). The results showed that the expression level of *SgGH3.1* in stylo leaves increased gradually as cold treatment extended, and peaked after 24-h cold treatment, when the expression level of *SgGH3.1* was 6.6 times of that before treatment (Figure 2B).

### 3.3. Overexpression of SgGH3.1 Altered IAA Sensitivity in Arabidopsis

Three *SgGH3.1* overexpressing *Arabidopsis* lines—OE1, 3, and 4—were obtained and verified by basta selection and successful amplification of *bar* and *SgGH3.1* genes (Appendix A). WT and the T3 transgenic lines were cultured in MS medium supplemented with 0, 2, 4, 6, 8, and 10 μM IAA (Appendix A). IAA exhibited different effects on the growth of underground and aboveground parts of *Arabidopsis*. As IAA concentration increased, the growth of *Arabidopsis* root was significantly inhibited, while the leaf length and width increased until IAA concentration reached 10 μM (Figure 3). No difference was observed between WT and *SgGH3.1*-OE lines when no IAA was added to the MS medium. The differences began to show when IAA was added, and became most significant when IAA concentrations reached 6 and 8 μM. The root length and leaf length of *SgGH3.1*-OE lines were significantly larger than WT when cultured in medium with 6 and 8 μM IAA.

### 3.4. Overexpression of SgGH3.1 Enhanced Chilling and Cold Tolerance in Arabidopsis

WT and *SgGH3.1*-OE *Arabidopsis* were seeded on MS medium and cultured at 22 °C and 10 °C. There was no significant difference in growth between WT and *SgGH3.1*-OE lines at 22 °C. When grown at 10 °C, the growth of WT and SgGH3.1-OE lines were all considerably slowed and their leaves grew slimmer and brittle. In comparison to the *SgGH3.1*-OE lines, WT has significantly smaller rosettes and leaves, as well as shorter roots, indicating that its growth was more hampered by chilling stress (Figure 4).

Three-week old *SgGH3.1*-OE and WT *Arabidopsis* seedlings were exposed to cold treatment at −4 °C (12 h dark/12 h light). After 2 weeks of recovery at 22 °C, WT leaves showed obvious symptoms of wilting, yellowing and mortality, whereas most *SgGH3.1*-OEs survived and their leaves, albeit thin and brittle, stayed green (Figure 5A), and the survival rates of *SgGH3.1*-OEs were significantly higher than WT (Figure 5B). To further investigate the improved cold tolerance of *SgGH3.1*-OEs, stress-related physiological changes and *CBF1-3* gene expressions were determined in *SgGH3.1*-OEs and WT when subjected to cold stress. There was no significant change in any physiological indices between *SgGH3.1*-OEs and WT under normal growing conditions. All physiological indices were increased in both SgGH3.1-OEs and WT when they were subjected to cold treatment, although no significant difference was found at first between SgGH3.1-OE and WT plantlets (after 2-h cold treatment). With the cold treatment continued, *SgGH3.1*-OE started to show enhanced physiological responses to low temperature compared to WT, including significantly higher proline, soluble sugar and soluble protein contents, and significantly lower MDA content and relative electrolyte leakage, and the differences between *SgGH3.1*-OEs and WT grew as treatment continued (Figure 5B). Before cold treatment, *CBF1*, *2*, and *3* gene expressions were significantly higher in two *SgGH3.1*-OE lines than in WT. After one hour of cold treatment, the expression levels of *CBF1*, *2*, and *3* were significantly upregulated to 18–92 times of the original levels, *CBF1* and *3* expression levels in *SgGH3.1*-OE and WT were not significantly different, while *CBF2* expression levels were 3.46 and 2.61 times higher in *SgGH3.1*-OE1 and *SgGH3.1*-OE3 than in WT, respectively (Figure 5C).

## 4. Discussion

Expression of some *GH3* genes has been proven to be tissue or organ-specific and responsive to biotic and abiotic stresses in many plant species, including *Arabidopsis* (*A. thaliana*), rice (*Oryza sativa*), maize (*Zea mays*), tomato (*Solanum lycopersicum*), chickpea (*Cicer arietinum*), soybean (*Glycine max*), *Medicago* (*Medicago truncatula*), *Lotus* (*Lotus japonicus*), and oilseed rape (*Brassica napus*) [15,18,19,28,29,30,31]. In this study, expression of *SgGH3.1* was detected in tissues of leaf, stem, and root of fine-stem stylo. Since leaves could rapidly sense low temperature and were easier to sample, we chose leaves to determine expression pattern of *SgGH3.1* under cold stress. The results showed that *SgGH3.1* was quickly induced by low temperature, suggesting that *SgGH3.1* is a cold-responsive gene and may play a role in cold responses of fine-stem stylo.

In plants, endogenous auxins were maintained at appropriate levels through regulation of auxin biosynthesis and distribution among different organs, and through conjugation of auxins with sugars, peptides, and amino acids (18–20). Although the physiological function of conjugates in auxin homeostasis is not yet fully understood, it is generally accepted that conjugate formation is crucial to auxin action. According to the phylogenetic tree, *SgGH3.1* belongs to the group Ⅱ of the GH3 family (Figure 1A). Members of this group have similar substrate preference, specifically conjugate auxins with amino acids in vitro and in vivo. As a result, we hypothesized that SgGH3.1 has the ability to conjugate IAA, like its homologs, and thus plays a role in IAA homeostasis. If the hypothesis is correct, *SgGH3.1*-OEs will react to exogenous IAA differently compared to WT. Here, we compared the growth of WT and *SgGH3.1*-OE *Arabidopsis* under different concentration of exogenous IAA. The responses of the aboveground and underground parts to IAA were completely opposite. The root growth was significantly inhibited even when only 2 μM IAA was added, while the leaf elongation was promoted by IAA until the concentration reached 10 μM. When IAA was not added to the growth medium, no severe morphological abbreviation was observed in *SgGH3.1*-OE plants compared to WT, but as IAA concentration increased to 6 and 8 μM, root and leaf lengths in *SgGH3.1*-OE lines were significantly longer than in WT (Figure 3). The results indicated that overexpression of *SgGH3.1* conferred higher resistance to IAA suppression of root growth and enhanced IAA stimulation of leaf elongation in *Arabidopsis*. Similar results were observed in a previous study. An *Arabidopsis* mutant *wes1-D* that overexpresses *AtGH3.5* was slightly less sensitive to IAA inhibition of primary root growth, whereas a T-DNA insertional mutant *wes1* was more sensitive [19]. We speculated that overexpression of *SgGH3.1* increased IAA conjugation and decreased endogenous IAA content, but not sufficiently to cause significant changes in root and leaf growth in *SgGH3.1*-OEs. When 2 μM IAA was applied, root length in *SgGH3.1*-OEs were inhibited more than in WT, which could be due to the overexpressing lines having a lower content of endogenous IAA and therefore being more sensitive to exogenous IAA. However, when the exogenous IAA concentration was increased to 6 and 8 µM, the overexpressing lines with the help of *SgGH3.1* could more effectively conjugate excess IAA than WT, which makes them more tolerant to the inhibitory effect of exogenous IAA on root growth. Because leaves are less sensitive to IAA than roots, when root growth is inhibited, leaf elongation is promoted, but when IAA concentrations exceed the plant’s regulatory threshold (10 µM in our study), leaf growth is also inhibited. When exogenous IAA levels are within a certain range (6 and 8 µM), IAA stimulated leaf elongation more strongly in *SgGH3.1*-OEs, which could be due to SgGH3.1′s ability to control IAA levels, allowing *Arabidopsis* to better maintain auxin homeostasis for leaf elongation. However, it should be noted that the morphological differences between WT and *SgGH3.1*-OEs were not as significant, which could be due to the following factors: (1) Because *SgGH3.1* was heterologously expressed in *Arabidopsis*, it might have a restricted regulatory function in IAA homeostasis. (2) At the time of measurement, seedlings were relatively small, and WT and *SgGH3.1*-OEs phenotypes were not highly distinct, meanwhile inconsistent germination and growth led to relatively large variations of morphological data within the same group, making the differences less significant as well. To corroborate this conclusion, we need to further refine the exogenous IAA treatment and morphological determination, as well as examine the difference of endogenous IAA contents between WT and *SgGH3.1*-OEs in the future.

When plants are under stress conditions, reallocation of metabolic resources between different physiological pathways will occur, leading to stressed symptoms including growth retardation and reduced metabolism. Auxin has been demonstrated to involve with such adaptive responses [32,33,34]. In this study, *SgGH3.1*-OEs and WT plants germinated and grew at 10 °C showed different levels of growth retardation, and the symptoms of WT were much more severe than those of *SgGH3.1*-OEs. We speculated that chilling stress (10 °C) would induce IAA synthesis and accumulation, and therefore inhibit plant growth, and that overexpression of *SgGH3.1* would partially counteract this inhibitory effect through IAA conjugation.

Group Ⅱ GH3 proteins play crucial roles in biotic and abiotic stress responses. Transgenic rice lines over-expressing *OsGH3.1* exhibit inhibited cell growth and cell wall loosening, as well as enhanced resistance to fungal pathogen due to reduced auxin contents [14]. OsGH3.2 differentially affects cold and drought tolerance in rice through modulating both endogenous free IAA and ABA homeostasis [15]. Transgenic rice lines overexpressing *OsGH3.8* exhibit reduced auxin contents, leading to altered plant growth and development and enhanced pathogen resistance [16]. Transgenic rice lines overexpressing OsGH3.13 also exhibit reduced auxin contents, leading to enhanced drought tolerance and altered root morphology [17]. Overexpression of GH3.3, GH3.5, GH3.6 and GH3.12 in rice individually improves rice resistance to bacterial pathogen Xoo through modulating JA homeostasis and regulating expression of JA-responsive genes [18]. An *Arabidopsis* mutant, *wes1-D*, in which a *GH3* gene *WES1/GH3.5* is activated by nearby insertion of the 35S enhancer, showed enhanced resistance to both biotic (pathogen infection) and abiotic stresses (including cold, heat, and drought stresses), and stress-responsive genes, such as pathogenesis-related genes and *CBF* genes, were up-regulated in this mutant [19]. Overexpression of *CsGH3.1* and *CsGH3.1L* in Wanjingcheng orange (*Citrus sinensis* Osbeck) reduces plant susceptibility to citrus canker by repressing auxin signaling and enhancing defense responses. Here, we also found that overexpression of *SgGH3.1* could up-regulate the expression of *CBF1-3* genes and activate the physiological responses against cold stress, therefore enhancing cold tolerance in *Arabidopsis*. CBF is also known as the dehydration-responsive protein-binding factor (DREB), and CBF1-3 have been identified as the most important transcription factors in the regulation of cold response gene (*COR*) expression, and essential for plant cold acclimation and cold resistance [10,11,35,36,37,38]. Many studies have demonstrated that mutation or altering the transcript level of the *CBF1-3* genes could dramatically change cold tolerance in plants [10,11,37,39,40,41]. Park et al. (2007) demonstrated that the transcription of *AtCBF1-3* was repressed by exogenous IAA in a concentration-dependent manner, and the expression of *AtCBF1* and *2* was increased in a *WES1/AtGH3.5* overexpressing mutant of *Arabidopsis*, *wes1-D* [19]. Thus, it is likely that *SgGH3.1* overexpression lowered endogenous IAA concentration, leading to increased *CBF1-3* expression in *SgGH3.1*-OE lines, which aided *Arabidopsis* in cold acclimation.

When a plant is exposed to cold stress, a number of genes are activated, resulting in elevated levels of a series of metabolites and proteins, some of which may be responsible for giving some degree of cold tolerance. Understanding the changes in cellular, metabolic, and molecular machinery that occur in response to cold stress has been critical to progress in breeding better crops under cold stress, which in turn gives new tools and techniques to enhance cold tolerance in crops. Low-temperature restrictions have been overcome by the discovery of cold-tolerant genes for use in genetically modified crops [42,43,44,45,46]. Transgenic technology has made it possible to improve cold tolerance in plants by introducing or removing a gene or genes that govern a specific characteristic [47]. It also provides unusual chances for improving plant genetic potential through the production of specialized crop varieties that are more resistant to cold stress. Because many aspects of the cold adaptation process are under transcriptional control, many transcription factors were chosen; thus, genetic engineering for introgression of such genes that are known to be involved in cold stress response and putative tolerance, may prove to be a faster track towards improving crop varieties for enhanced cold tolerance. Although *SgGH3.1* is not a transcription factor, it can influence the transcription of AtCBF1, 2, and 3, the key regulators of the cold responsive network, and increase *Arabidopsis* chilling and cold tolerance. Although the specific mechanism of *SgGH3.1*′s effect on the cold responsive pathway and low-temperature tolerance remains unclear, it has shown great potential as a candidate gene for cold tolerance breeding. Hopefully, in the near future, when stylo transgenic technology is fully established in our lab, we will be able to generate new variety with enhanced cold tolerance by overexpressing or editing *SgGH3.1*.

## 5. Conclusions

The results taken together, *SgGH3.1* encodes as a Group Ⅱ GH3 protein consisting of 602 amino acids. *SgGH3.1* was expressed in root, stem, and leaf tissues, and responds promptly to cold stress in leaves. Overexpression of *SgGH3.1* in *Arabidopsis* has changed sensitivity to exogenous IAA, up-regulated *AtCBF1-3* transcriptions, and conferred significantly improved chilling and cold tolerances to the transgenic plants. Our findings confirmed *SgGH3.1*′s involvement in IAA homeostasis and cold responses, and provides insights for the improvement of cold tolerance in stylos.

## Figures and Tables

**Figure 1 genes-12-01367-f001:**
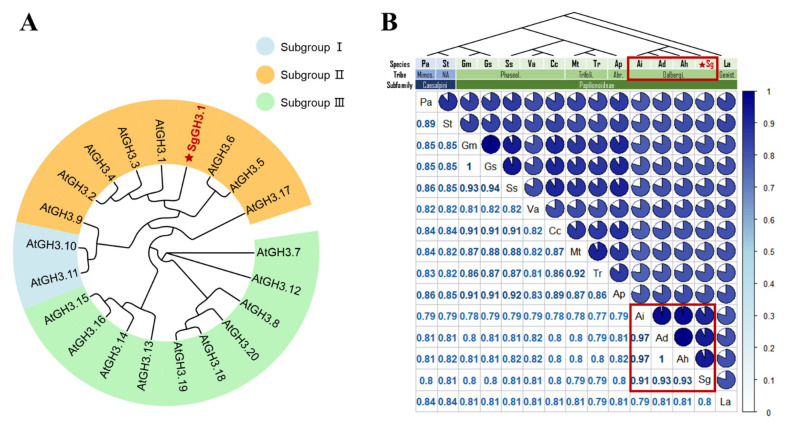
Phylogenetic tree of *SgGH3.1* and *Arabidopsis* Gretchen-Hagen 3 (GH3) proteins (**A**) and percentage identity matrix of *SgGH3.1* and GH3.1 homologous proteins of 14 leguminous species (**B**). T-Coffee was used for the multiple alignment and MrBayes for the tree construction. The light blue, orange, and light green indicate Subgroup Ⅰ, Ⅱ, and Ⅲ of GH3 family, respectively. The pie chart on the upper right and the numbers on the lower left of the matrix represent the percentage identities between the leguminous GH3.1 protein sequences. The blue color of the pie charts and numbers indicates the level of the percentage identity (the darker the blue, the higher the percentage identity). The abbreviations on the diagonal line represent different leguminous species: Pa for *Prosopis alba*, St for *Senna tora*, Gm for *Glycine max*, Gs for *Glycine soja*, Ss for *Spatholobus suberectus*, Va for *Vigna angularis*, Cc for *Cajanus cajan*, Mt for *Medicago truncatula*, Tr for *Trifolium repens*, Ap for *Abrus precatorius*, Ai for *Arachis ipaensis*, Ad for *Arachis duranensis*, Ah for *A. hypogaea*, Sg for *S. guianensis*, and La for *Lupinus albus*. The tree above the matrix is based on the phylogenetic analysis of the 15 leguminous GH3.1 protein sequences.

**Figure 2 genes-12-01367-f002:**
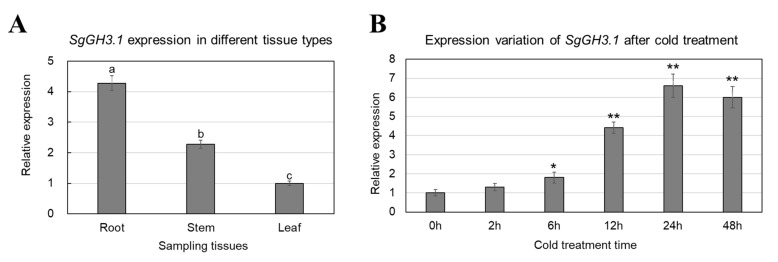
Expression of *SgGH3.1* in different tissues of stylo (**A**) and expression variation of *SgGH3.1* in leaf tissues of stylo after cold treatment (**B**). Different letters above the columns indicate significant differences at *p* < 0.01 between different tissues. * and ** indicate significant differences at *p* < 0.05 and *p* < 0.01 with 0 h, respectively.

**Figure 3 genes-12-01367-f003:**
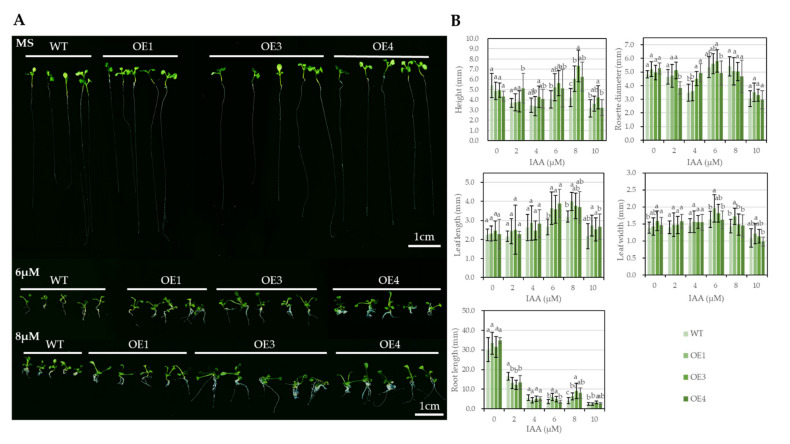
Phenotypic images and measurements of wild-type (WT) and *SgGH3.1* overexpressing (OE1, 3 and 4) *A. thaliana* seedlings grown in MS medium supplemented with different concentrations of IAA. (**A**) Scans of WT and *SgGH3.1*-OE *A. thaliana* seedlings grown in MS medium supplemented with 0, 6 and 8 μM IAA. (**B**) Height, rosette diameter, leaf length and width, and root length of WT and *SgGH3.1*-OE *A. thaliana* seedlings grown in MS medium supplemented with 0, 2, 4, 6, 8 and 10 μM IAA. Different letters above the columns indicate significant differences at *p* < 0.01 between different IAA treatments.

**Figure 4 genes-12-01367-f004:**
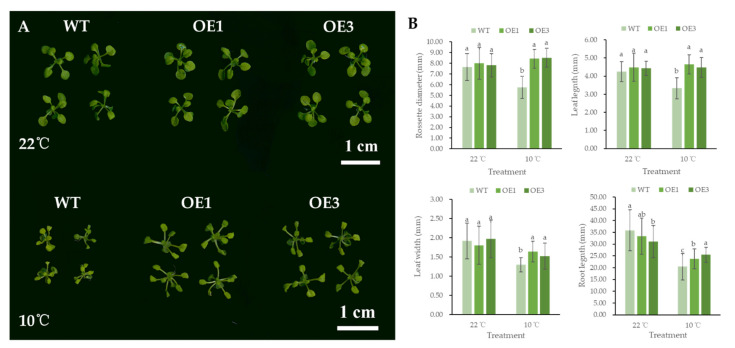
Phenotypic differences between wild-type (WT) and *SgGH3.1* overexpressing (OE1 and 3) *A. thaliana* seedlings cultured at 22 °C and 10 °C. (**A**) Scans of *Arabidopsis* seedlings. (**B**) Phenotypic measurements of *Arabidopsis* seedlings. Different letters above the columns indicate significant differences at *p* < 0.05 between different *Arabidopsis* lines within the same treatment.

**Figure 5 genes-12-01367-f005:**
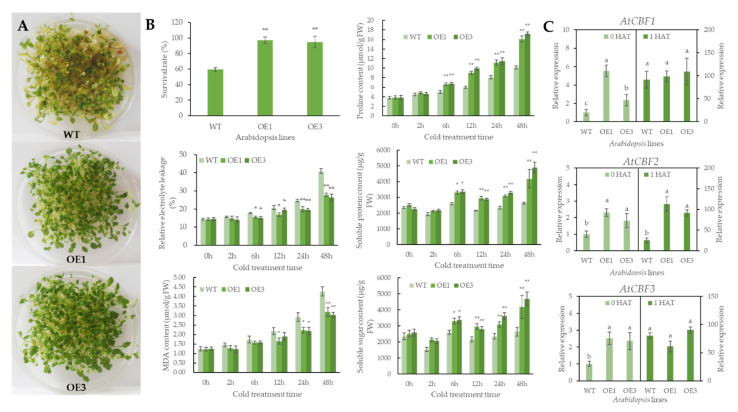
Comparison of growth, physiological indices and *CBF* gene expressions between wild-type (WT) and *SgGH3.1* overexpressing (OE1 and 3) *A. thaliana* seedlings after cold treatment. (**A**) Growth of WT and *SgGH3.1*-OE lines after 2-week recovery following cold treatment. (**B**) Survival rates and physiological variations of WT and *SgGH3.1*-OE lines after cold treatment, * and ** indicate significant differences at *p* < 0.05 and *p* < 0.01 with 0 h, respectively. (**C**) Expression variations of *AtCBF1, 2 and 3* genes in WT and *SgGH3.1*-OE lines after 1 h cold treatment, the left Y axis represents the expression level prior to cold treatment (0 HAT), and the right Y axis represents the expression level 1 h after cold treatment (1 HAT), different letters above the columns indicate significant differences at *p* < 0.05 between different *Arabidopsis* lines within the same treatment.

## Data Availability

The data presented in this study are available on request from the corresponding authors.

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
