# Peer review of "Overexpression of SgGH3.1 from Fine-Stem Stylo (Stylosanthes guianensis var. intermedia) Enhances Chilling and Cold Tolerance in Arabidopsis thaliana"

_genes, 2021, doi:10.3390/genes12091367_

Round 1

Reviewer 1 Report

Ressults presented by Authors is important and original. However, several serious flaws should be corrected. Below is the precise description of suggested changes:

Introduction

Authors wrote of biological functions of CBF proteins in discussion section- it is ok. However, some sentences concerning role of CBF (particularly in regulation response to auxin or cold stress) could be provided in the introduction fragment.

Authors could also add several sentences related to the typical physiological changes after cold treatment as MDA, soluble sugar, soluble protein and proline concentration or electrolyte leakage. Authors could shortly describe role or mechanism of these factors in plant response to cold stress  

Section 2.1

Details of PCR reaction used to obtain cDNA of SgGH3.1-temperatures and duration of PCR reaction stages

Used DNA polymerase, amount of DNA polymerase per PCR reaction

Was the PCR product cloned into a cloning vector for further transfer into the plant overexpressing vector pBA002? For example using TA/TOPO TA or any other common PCR product cloning method?

Section 2.2

How the homozygous T3 transgenic lines were prepared- for example selfing by 6-8 generations or  culturing of haploid cells (anther or pollen) that were then converted into diploid cells or any other method used by Authors. Provide method description plus citation- if possible.

Section 2.6

Assessment of RNA purity, concentration and integrity

Details of DNase treatment to remove putative remnants of genomic DNA

Reverse transcription reaction; details of reaction-temperature and time, volume and amount of used RNA

qPCR target- length of PCR product (control-actin and tested gene), target gene symbol and accession number,

qPCR protocol: conditions of PCR reaction, volume of reaction, concentration of magnesium ions, dNTPs, DNA polymerase type and concentration.

Citation of method used to calculate RT-PCR results- for example Livak and Schmittgen 2001 or other.

Section 3.3

It is not clear if OE1,3 and 4 are T1 or T3 lines. It should be clearly stated in section 3.3 s well as  materials and methods section.

Figure 3B

Maybe more clear could be using * or ** as in Fig. 2B and 5B and captcha to Fig. 2B and 5B instead of letters a,b,c as it is in Fig 3B and captcha to it?

Figure 4B- the same comments as for Fig 3B

Discussion

Line 304:

Should be 35 S enhancer (or promoter), not 35S enhancer

Line 312- mistakenly provided abbreviation of CBF

CBF or better CBF/DREB1 should be: C repeat binding factor/dehydration-responsive protein-binding factor1. For example see following article:

The cold response regulator CBF1 promotes Arabidopsis hypocotyl growth at ambient temperatures | The EMBO Journal (embopress.org)

Reviewer 2 Report

The paper from Jiang et al describes the overexpression of the Stylo GH3.1 gene in Arabidopsis thaliana and its effect on cold response. The authors first confirmed that SgGH3.1 effectively belongs to the GH3 family through a phylogenetic analysis and further studied the expression of SgGH3 in Stylo. They studied the response to IAA of Arabidopsis plants overexpressing SgGH3.1 in order to confirm the involvement of this gene in auxin homeostasis. Finally, they studied the morphologic and physiologic responses of the transgenic lines to cold stress as compared to the WT. The paper is generally well-written, the objective is clear and the methods used are appropriate to support the results. I would however like to request more precise description of the methods (see below). Furthermore, I have some concerns regarding the interpretation of some data by the authors, especially concerning the leaf phenotypes in the transgenics in response to IAA and the expression of CBF1-3 in response to cold stress (see below). Generally speaking I feel the paper lack essential data that are needed to support the conclusions as qPCR results showing the overexpression of GH3 in transgenics as compared to WT, morphological results for the cold experiment, or measurements of IAA content in response to cold in the transgenic lines as compared to WT for instance (author may already have these data actually).

Methods: please provide a description of the statistics used for analysis.

Fig 3B and 4B: in view of the error bars, some significant differences appear surprising. Please double check and provide a description of the statistical test that were used.

Fig3B: overall differences in leaf length across the different IAA concentration are around 1mm. That looks marginal and conclusion related to these results seem not robust (L264-266).

Fig 5A: Based on this Figure, I am not convinced that cold symptoms are less severe in the transgenic lines. Quantitative data would be useful. I am curious why authors waited for 2 weeks to observe the recovery symptoms and why they did not measure the physiological indices at that time.

Fig5C: I am absolutely not convinced by the differences described by authors in AtCBF1, 2 and 3 expression in the transgenics as compared to the WT. To me the link between CBF and cold response (stated in L310-312) is very difficult to make since there is no induction or inconsistent induction of their expression in response to cold. L223-226: comparison should be made between treatments for the same genotypes rather than between genotypes I believe.

Reviewer 3 Report

I have following suggestion.

1) I will suggest to perform the over-expression of SgGH3.1 gene in Stylo and observe the phenotype. Compare the results with over expression lines of Arabidopsis.

2) Mutate the gene SgGH3.1 gene in Stylo and observe the phenotype. 

3) If over-expression and mutant analysis experiments in Stylo goes well and if you observe enhancement in chilling and cold tolerance in Stylo, we can conclude that SgGH3.1 gene is really enhances the chilling and cold tolerance. 

Till then, additional experiments are needed. Its hard to conclude experiment based on current results.

Round 2

Reviewer 1 Report

Dear Authors,

Thank you for comments that addressed questions presented in my report. They significantly improved the manuscript. I have no other comments related to the manuscript.

Reviewer 2 Report

This second version of the manuscript has improved. Authors have taken into considerations some weaknesses of their work and tried to discuss them. The readers are now aware, but the concerns regarding the validity of some conclusions remain. 

Reviewer 3 Report

The authors has addressed the points raised by me. Manuscript can be accepted in present form.